# Link of Patient Care Outcome to Occupational Differences in Response to Human Resource Management: A Cross-Sectional Comparative Study on Hospital Doctors and Nurses in China

**DOI:** 10.3390/ijerph17124379

**Published:** 2020-06-18

**Authors:** Chaojie Liu, Timothy Bartram, Sandra G. Leggat

**Affiliations:** 1School of Psychology and Public Health, La Trobe University, Melbourne VIC 3086, Australia; s.leggat@latrobe.edu.au; 2School of Business, RMIT University, Melbourne VIC 3000, Australia; timothy.bartram@rmit.edu.au

**Keywords:** high performance work systems, quality of care, health professionals, health services management, China

## Abstract

This study assessed the link of patient care outcome to occupational differences in response to human resource management. A cross-sectional survey was conducted in three large regional hospitals in China. A total of 700 questionnaires were distributed to doctors, nurses, allied health workers, and managers and 499 (71%) were completed. Data were analyzed using a final sample of 193 doctors and 180 nurses. Quality of patient care was rated by the participants using a modified version of the Victorian Patient Satisfaction Questionnaire. Human resource management was measured on seven aspects: job security, recruitment, training, transformational leadership, information sharing, job quality, and teams. The differences between doctors and nurses in response to the human resource management practices and their associations with quality of care were compared through independent samples *t*-tests, correlational analyses, and moderator regressions. Doctors gave a higher rating on quality of patient care than their nurse counterparts. ‘Training’, ‘transformational leadership’, and ‘information sharing’ were more likely to be associated with higher ratings on quality of patient care in nurses. By contrast, a greater association between ‘teams’ and quality of patient care was found in doctors. Although doctors and nurses in China are exposed to the same hospital management environment, professional differences may have led them to respond to management practices in different ways.

## 1. Introduction

In the business sector, a series of management practices have been identified for motivating staff to achieve organizational goals and improve performance. Bundles of such management practices were labeled as high-performance work systems (HPWS) [1]. While researchers propose different configurations of HPWS, there is consensus that HPWS comprise components of management practice that influence and align employees’ attitudes and behaviors with the strategic goals of the organization and thereby increase employee commitment and ultimately individual performance [2]. The HPWS management practices according to the international literature [3,4,5], including China [6,7], comprise: job security; selective hiring; opportunities for training; job quality; transformational leadership; information sharing; and the use of teams.

Over the past two decades, there has been growing interest in the health sector about the role of HPWS in promoting quality patient care [4,7,8,9]. Although there is little doubt about the need to put quality at the center of patient care, healthcare organizations are often subject to many competing pressures deriving from multiple stakeholders, notably consumers, third party payers (governments or insurance funds), and healthcare professionals. Unfortunately, these pressures are not always well aligned with the interest of patients. Aiken and colleagues [10] warn, in a study published in the Lancet, that austerity measures imposed by governments such as reduction of nursing staff numbers can result in significant loss of life of hospital patients. Despite strong empirical evidence supporting the association between work environment and quality of patient care [11], our understanding about how management environments shape health professional practices is quite limited. Recently, some researchers pointed out that HPWS may have a “dark side” in terms of increasing stress levels of employees [12,13,14,15]. This raises concerns about its potential deleterious health effects not only on health workers as employees but also on patients they serve [16].

Health care organizations are extremely complex, employing highly skilled professionals with high autonomy in decision making. Seminal work by Lepak and Snell [17] demonstrated that different occupational groups, with their own history, occupational values, work structures, and employment relationships may view management in different ways. Bartram et al. [18] found that perceptions of HPWS were lower for medical and support service workers as compared with nurses and management/administrative staff in a large Australian hospital. If it is true that nurses and doctors respond to the same management practice in different ways, this could have implications for how they work together in delivering patient care. Team effectiveness is dependent on each team member interpreting management practices in the intended way—that is—all team members ‘singing the same song’ [19,20]. This need for coherence is supported by Gittell et al.’s assertion that HPWS work through relational coordination [21].

Scholars have called for research analyzing responses by different employee groups within the same organizational settings [22]. However, the majority of healthcare studies have tended to regard hospital employees as an homogenous group of workers [1], or have focused on single occupational groups [23]. Health professional groups, particularly doctors and nurses, have a long history of their own traditions, values, and professional practices. According to McGivern et al. [24], a “professional is an exclusive identity, developed through qualifications, training, and socialization, creating social identity boundaries and enhanced careers”. Professions use a process of socialization, which develops collective understandings of the purpose, values, and culture of the professional group, which can then be used to provide a framework for individual meaning [25]. This has long been recognized as creating norms of loyalty and service [26] and is particularly relevant for doctors and nurses [27]. A major challenge in hospitals is that of the competing logics of health professionals, as they create ‘countervailing determinants of power and bring rivalry to the fore’, especially between doctors and nurses [28].

This study explores whether doctors and nurses working in Chinese hospitals have different responses to the hospital management practices, measured as HPWS, and their links to the quality of care provided. Hospitals in China offer an ideal setting for such a study. Unlike many Western countries where doctors often work across several facilities and are exposed to different management arrangements (e.g., senior doctors who are consultants rather than employees), Chinese doctors and nurses usually seek permanent employment with one hospital employer over their career [6].

The current study aims to explore the links between patient care outcomes and occupational differences between doctors and nurses in response to human resource management practices used in Chinese hospitals. The paper makes two contributions to the literature. First, the study advances our understanding of the impact of HPWS on patient care by unpacking the components of HPWS and comparing the differences in the association between each HPWS component and quality of patient care. Most studies on HPWS management practices aggregate the construct and do not examine the effect of individual components on employee attitudes and behaviors. Ang and colleagues [1] argued that it is very important to examine how health workers respond to different components of HPWS in different ways. This is simply because health professionals are in a prestigious powerful position [29], which plays a pivotal role in any change in hospitals that could impact on the nature and process of patient care [30]. Second, through comparing the differences between doctors and nurses in response to different components of HPWS, the study enriches the knowledge related to how different occupations should be managed differently [31]. Doctors generally decide which patients enter the hospital, leave, and what treatment they receive. They are trained to be independent, rely on their own judgment, and to be accountable to their profession rather than their organization [24]. In contrast, professionally trained nurses are usually working as a team by themselves and with doctors, providing the majority of care for patients. In the hospital setting, implementation of doctors’ advices and organizational procedures heavily depends on nursing services. In theory, the differences in professional identity and norms are likely to result in different responses to HPWS and consequently patient care outcomes [32].

## 2. Materials and Methods 

### 2.1. Study Setting

A cross-sectional paper-based questionnaire survey was undertaken in three public hospitals in China, with a sample of 193 doctors and 180 nurses. In China, most doctors and nurses in public hospitals are full-time employees that have been exposed to the same management environment. The Chinese culture encourages a collective approach towards professional work; but also values hierarchy with clear status distinctions [33]. This has been accompanied by growing medical dominance and professional autonomy, leading to enlarged status distinctions between doctors and nurses. It is evident that health sector reforms in China have impacted doctors and nurses in different ways [34]. This presented an opportunity to explore the differences of doctors and nurses in responding to the same management practices.

Although nurses in China do not enjoy the same level of training and status as nurses in other countries, nursing as an independent profession has grown stronger in the past few decades. There has been an increasing awareness of the importance of professional nursing services in patient safety and quality of care. A nurse registration system was set up in parallel with the medical registration system, which has been accompanied with increasing numbers of university nursing courses [35]. This has also led to recognition of the importance of human resources management for attraction and retention of nurses, such as improving nursing work cultures for retention, investing in nursing education, and enhancing the image of the profession [36]. However, in contrast to Western nursing practice, the scope of nursing practice in China has largely remained restricted to traditional roles, with a high degree of dependence on instructions from medical doctors. The lack of commitment to nursing as a professional career in China has attracted a great deal of concern [35]. 

The juxtaposition of the recognition of the importance of HPWS in relation to the Chinese health sector reforms and traditional doctor–nurse professional relationships led to the development of this study. In the move from a developing to a developed country, China’s health policy has modernized its health care system, with more doctors and nurses obtaining university level education. However, there is still a large proportion of the nursing workforce with secondary school level training [37]. As a result, the practice of nursing is largely task oriented [34]. In contrast, doctors have increased their status and power, to more closely resemble the medical workforce in Western countries. In fact, the early waves of reforms reduced government control over medical practitioners to a greater extent than seen in Western countries [38]. The existing hospital system is largely decentralized and market-driven, and the quality of care initiatives found in other countries may not be compatible with the financial incentives that doctors and hospitals in China currently enjoy [38]. With decreased budget support from the government, hospital revenues have become increasingly dependent on user charges for services. This enabled doctors to operate as independent businesses, creating their own demand and generating their own incomes through sales of medications and other treatments [39].

### 2.2. Sampling and Participants

This study was undertaken in three regional hospitals: two in Beijing with 653 and 1160 beds, respectively, and one in Tsingtao with 750 beds. They are all general public hospitals, providing a full range of medical and surgical services. Hospital employees were invited to complete the questionnaire, with 700 questionnaires (equivalent to 15% of hospital employees) distributed to all clinical units and service departments in 2011 and 2012. Medical doctors, nurses, allied health workers, and managers were invited to complete a paper-and-pencil questionnaire survey without supervision. Participants were required to read the informed consent letter on the cover page before proceeding to the questionnaire. Return of the questionnaire was anonymous and voluntary. Respondents could withdraw at any stage, complete only part of the questionnaire, or return an empty questionnaire to the collection box. The study protocol was approved by the human research ethics committee of La Trobe University.

A total of 499 (71%) questionnaires were completed. Respondents included in this study were those who identified themselves as a doctor or a nurse. This resulted in a sample size of 373, including 193 doctors and 180 nurses. The respondents had worked in the hospitals for a median of 10 years (interquartile range 5–20 years), with a mean workload of 45.6 h (SD = 8.5 h) per week. On average, nurses reported a lower (*p* < 0.01) workload (41.9 ± 4.5) than doctors (49.5 ± 9.8).

### 2.3. Measures

The questionnaire included five constructs: (1) human resources management environment, (2) quality of patient care, (3) psychological empowerment, (4) affective commitment, and (5) trust. These measurements were selected based on the existing body of HPWS studies. Reported psychological empowerment, affective commitment to the organization, and organizational trust have been found to mediate the relationship between HPWS and individual and organizational outcomes [6,19]. It is evident that occupational differences in psychological empowerment, affective commitment, and organizational trust exist in the health sector as a result of its hierarchical structure and the varied nature of employer–employee relationship [7]. A participatory approach in management decisions may alleviate some of the negative effects of the hierarchical inter-professional arrangements in healthcare services [6].

The experience of the respondents with management environment was measured using a validated HPWS scale that included job security, recruitment, training, transformational leadership, information sharing, job quality, and teams [40]. The components were each measured with two to eight items on a five-point Likert scale. For example, job security was measured by two items: ‘Providing employment security to employees is a priority in this unit’ and ‘If an employee were to lose his/her job, this unit would try very hard to find him/her another position elsewhere in the organization’. Transformational leadership was measured by seven items. Example items are ‘My immediate manager is focused on doing the right thing, as well as on getting results’ and ‘My immediate manager provides employees with continuous encouragement’. Teams was measured by four items, such as ‘My unit supports team development and training’ and ‘The development of teams is an important element for my unit’. The scales had a Cronbach’s α coefficient ranging from 0.677 to 0.939, indicating acceptable or good internal consistency.

The measurement of quality of care has not been easy in healthcare occupational studies. It is often difficult to get patient level quality of care data that can be meaningfully correlated with organizational and staff psychological constructs and attitudes. To overcome this, staff ratings on the quality of patient care have been shown to act as a useful proxy for patient level indicators [41] and were adopted in this study using a modified version (16 items) of the Victorian Patient Satisfaction Questionnaire [23]. Example items are ‘I respond to patients/clients call quickly’ and ‘I treat patients/clients with respect’. Each item was rated on a five-point Likert scale. The scale had a Cronbach’s α coefficient of 0.823, indicating good internal consistency.

Psychological empowerment was measured using Spreitzer’s 12-item measure [42]. It contains four components, including competence (e.g., I have mastered the skills necessary for my job), impact (e.g., My impact on what happens in my workplace is large), meaning (e.g., The work I do is very important to me), and self-determination (e.g., I can decide on my own how to go about doing my work). Each component was measured by three items on a five-point Likert scale. The scale had a Cronbach’s α coefficient of 0.865, indicating good internal consistency.

Affective commitment to the organization was measured by four items adapted from Allen and Meyer [43]. Example items are “I would be very happy to spend the rest of my career with this organization” and “I really feel that this organization’s problems are my own”. Each item was rated on a five-point Likert scale. The scale had a Cronbach’s α coefficient of 0.628, indicating acceptable internal consistency.

The interpersonal trust at work scale [44] explored the participant’s level of trust in the organization and with the people they work with. Questions included “Management can be trusted to make sensible decisions for the firm’s future” and “I have full confidence in the skills of my workmates”. Each question item was rated on a five-point Likert scale. The scale had a Cronbach’s α coefficient of 0.856, indicating good internal consistency.

### 2.4. Statistical Analyses

The study compared doctors’ and nurses’ experiences of the eight component HPWS practices, and their ratings on quality of patient care. A summed score was calculated for each of the measurements. We compared the differences between doctors and nurses in relation to each of the measurements using independent sample *t*-test analysis. A correlational analysis between ratings on quality of patient care and the HPWS management practices was performed for the doctors and the nurses, respectively.

To further explore the impact of profession (doctors or nurses) on the association between ratings on quality of care and the management components, multivariate linear regression models were established with quality of patient care as the dependent variable. Seven models were developed, each containing one of the HPWS management components. The effects of profession were tested using a dichotomous variable “profession” (Doctor = 1, Nurse = 0) and an interaction covariate “profession*HPWS component”. The HPWS component scores were centered before interaction terms were introduced.

Control variables introduced into the regression models include sex (dichotomous), age (continuous), educational attainment (continuous), and work units (dichotomous). Previous studies revealed that psychological empowerment, organizational commitment, and trust in managers play an important role in the association between HPWS and performance [6]. They were also introduced into the regression models (as continuous variables) to reduce unobserved variances. Some of the questionnaire items contained missing values (up to 5). A casewise deletion strategy was adopted to handle the missing values.

Sensitivity tests were performed to test the robustness of the findings. The confirmation factor analysis (CFA) on the HPWS measurements with a seven-factor model showed that some items had low factor loadings (<0.4) on their respective latent variables. Deletion of these items improved and generated a good model fit. The regression analyses using variables exclusive of these items produced consistent results to those without exclusion of these items (Appendix A). We also performed partial least squares structural equation modeling (Appendix A), which confirmed the moderator role of profession on the association between the HPWS components and quality of patient care as revealed in the regression analyses. 

The data analyses were performed using SPSS/AMOS 25 and SmartPLS. A *p* value less than 0.05 was deemed statistically significant.

## 3. Results

### 3.1. Characteristics of Participants

The nurse respondents had a mean age of 33 years, 5 years younger than their doctor counterparts on average (*p* < 0.001). Over 97% of nurses were female, compared to 38% for doctors (*p* < 0.001). Doctors had a higher level of qualification than their nurse colleagues (*p* < 0.001): 55% of doctors held a postgraduate degree compared with 4% by nurses; by contrast, 49% of nurses held a certificate/associate degree compared with 8% of doctors. 

Higher levels of psychological empowerment (*p* = 0.003) and organizational commitment (*p* = 0.002) were found in doctors compared with nurses. No significant differences of trust (*p* = 0.83) appeared between doctors and nurses (Table 1).

### 3.2. Experiences of Management Environment and Patient Care Outcome

The doctor respondents gave a higher rating on quality of patient care (69.16 ± 6.75) than their nurse counterparts (65.98 ± 10.34, *p* = 0.001). However, their reported experiences on the HPWS management practices were similar (*p* > 0.10) (Table 2). Overall, a positive correlation between management practices and patient care was found in both doctors and nurses.

With respect to the individual HPWS components, ‘Job security’ and ‘Teams’ were found as significant factors associated with quality of patient care for doctor respondents, but not for nurse respondents. In contrast, ‘Information sharing’ was a significant factor associated with quality of patient care for nurse respondents, but not for doctor respondents. ‘Training’ and ‘Transformational leadership’ had a higher level of correlation with quality of patient care in nurse respondents compared with their doctor counterparts (Table 3).

### 3.3. Effects of Profession on the Impact of HPWS on Perceived Patient Care Outcomes

The regression models explained 30–38% of total variance of ratings on quality of patient care. The different ratings between doctors and nurses on quality of patient care were evident (*p* < 0.05) in all the regression models (Table 4). Profession was also found to moderate the effects of ‘Training’, ‘Transformational leadership’, ‘Information sharing’, and ‘Teams’ on quality of patient care. ‘Training’, ‘Transformational leadership’, and ‘Information sharing’ were likely to have a greater influence on nurses than doctors in relation to quality ratings on patient care. ‘Teams’, on the other hand, was likely to have a greater influence on doctors than nurses in relation to quality ratings on patient care. ‘Selective recruitment’ was not a significant predictor of quality ratings on patient care (Table 4). 

The structural equation modeling results showed that differences existed between doctors and nurses in the tested pathways between HPWS and patient care outcome ratings, especially over the path “HWPS -> Commitment -> Quality of Care” and the path “Trust -> Quality of Care”. Organizational commitment appeared to have a greater effect on doctors; whereas trust had a greater effect on nurses (Appendix A).

## 4. Discussion

### 4.1. Main Findings

This study found that there were no overall differences in the experiences of the doctors and nurses in relation to the existence of the aggregated HPWS practices. However, the doctors reported higher quality patient care than the nurses working in the same hospitals. Differences in the links between various HPWS components and quality of care appeared between doctors and nurses. Job security and teams were positively associated with quality of patient care from the perspective of doctors, whereas information sharing, training, and transformational leadership were more important from the perspective of nurses in relation to quality of patient care.

The data also suggested that the quality ratings on patient care were associated with the attitudes of doctors and nurses towards their workplace hospital. Doctors expressed significantly higher levels of psychological empowerment and organizational commitment than the nurses. These results are consistent with findings of previous studies [6,23].

To understand these phenomena, we need to use a complexity lens [45]. Differences in professional values and approaches may have shaped how doctors and nurses view patient care. But the underlying causes of such professional differences are usually deeply rooted in the context of health systems.

Throughout the world, the values of the medical profession promote an independent and individualist approach to care [27]. Doctors tend to focus on ‘fixing the immediate problem’ as their interactions with patients are episodic [46]. However, they have an expectation to depend on team effort in implementing their decisions. This may explain why a stronger association between teams and patient care was found in doctors and why organizational commitment had a greater effect on doctors. A previous study [47] has shown that doctors working on the same team may have wildly different perspectives of the quality of the teamwork. Generally, doctors hold more positive perceptions of the team than nurses, given they often lead those teams or serve as key decision makers.

By contrast, nurses tend to be subservient to doctors [35]. But they often work as a team and spend time with patients as the in-hospital care providers [46]. Studies have consistently shown that nurses strongly equate quality of care with the nursing organization and nursing practice environment [34]. This view is likely to be even stronger in China, as the evolution of the nursing profession has resulted in a predominantly task-focused nursing workforce. In comparison with other countries, a higher proportion of nurses in China rate the quality of their working environment as poor, which can translate into lower ratings of the quality of patient care [34].

The reform of the Chinese health care system has likely contributed to the professional differences between doctors and nurses. Empirical evidence shows that nurses in China have become increasingly subordinate to the doctors, not only in professional practice but also financially, which defies international trends and causes serious concerns [48]. In the late 1980s and early 1990s, China introduced market-oriented reform and government investment in public hospitals was reduced. Hospitals were encouraged to compete and generate revenues through user charges for services. This tied the hospital’s financial prospects to doctors’ performance. Doctors were rewarded not only for their professional expertise but also for their capacity to bring financial benefits to the hospital, resulting in reduced governance of medical practices. While there are moves underway to reign in the powers of the doctors [38] as would be expected, these doctors are unlikely to accept the need for these reforms, holding positive views of the care provided under the current system.

It is important to note that nurses and doctors experienced aggregated HPWS management practices in a similar way in this study. This is perhaps because doctors and nurses are exposed to the same strongly controlled management processes that China imposes in public hospitals. Management decisions, whether made through a participatory or non-participatory approach, have to be endorsed by government agencies. Local management decisions can be overruled if deemed incompatible with governmental ‘opinions and recommendations’ [49]. However, the exposure to the same HPWS management practices generated different psychological effects: nurses in this study reported significantly lower levels of psychological empowerment and organizational commitment than doctors, despite a similar level of trust in management. It is also important to note that trust was found to have a greater effect on nurses. Indeed, Chinese nurses do not work as equal team members [48]. Although the lower ratings of empowerment and commitment by the nurses, linked with their lowered status in the Chinese hospital system, may have a profound impact on their ratings on the impacts of teamwork, they do not offer a full explanation of the professional differences between doctors and nurses. The nurses identified mechanisms underlying successful teamwork, such as information sharing, transformational leadership, and training as having greater importance than teamwork itself. This is consistent with current views that in complex adaptive systems it is collaboration that is most important and teamwork is but one component of collaboration [45]. Even in the Chinese collectivist society, the perceptions of teamwork and its association with quality care were different among the doctors and nurses in our sample. We believe that the health system reforms, where doctors enjoy greater clinical autonomy and privileges than in Western public health systems, and where there are fewer nurses per capita, with an increasingly subservient role, have contributed to this divide.

Human resources management approaches are likely to have contributed to the development of occupational differences in professional culture between doctors and nurses and their responses to management practices. Status distinction is highly accepted in the health industry despite concerns of its potential detrimental effects on patient safety [6,50]. Nurses have a long history of working collegially with managers and supporting effective management policies and practices in hospitals [51]. However, doctors, with more independence of practice, have been less concerned with management policies and practices. It is important to note that the rising consumer movement and the increased acceptance of clinal governance by the medical profession are likely to encourage doctors to become more responsive to management policies and practices in the future.

### 4.2. Practice Implications

Hospital managers can enhance human resources management if they understand how doctors and nurses respond to HPWS management practices, and the association with quality of care. High quality health care is complex and requires health professionals to work together in interdisciplinary teams [7]. Given that different components of HPWS seem to have different effects on the quality ratings of patient care for doctors and nurses, we suggest that managers build on this understanding. For instance, consistent with the evidence [36], managers need to ensure nurses access HPWS practices such as information sharing, training, and transformational leadership.

Meanwhile, it is important to enhance the professional status of Chinese nurses. This is because inter-professional collaborations require “shared power based on knowledge, authority of role, and lack of hierarchy” [45]. The lower levels of empowerment and commitment perceived by nurses deserve increasing attention. The market driven hospital development in China has resulted in a dysfunctional hierarchical gap between hospital-based doctors and nurses, while health system reforms in Western countries aim to address the complexity and quality of hospital care [19]. A systems approach needs to be adopted to reduce the hierarchical gap involving changing policy, management, and educational arrangements [52]. These measures may eventually help win trust, a key pathway to improve quality of patient care from nurses.

### 4.3. Limitations

There are some limitations in this study. The study adopted a cross-sectional design and no causal relationships can be assumed. The study sample was restricted to several regional hospitals and may not be generalizable to other hospitals. We advocate further research into the effects of HPWS on inter-professional relationships and teamwork within the Chinese healthcare setting. One of the strengths of this study is the unpacking of the HPWS components, with measures derived from existing empirical studies. Unlike single measures for HPWS where a coherent theoretical framework is proposed [53], these multiple HPWS component measures have varied origins. Some of the measures may be more closely aligned with the formative assumption rather than the reflective assumption [54], in which case, the classical validation methods may not be appropriate. Further HPWS measurement development is warranted, as there is no one ‘correct’ HPWS measure.

## 5. Conclusions

Although doctors and nurses in China are exposed to the same hospital management environment, professional differences may lead them to respond to management practices in different ways. We have found that while Chinese doctors and nurses report similar experiences of management practices, the doctors in our sample displayed greater levels of empowerment and commitment, and also gave higher ratings on quality care delivery than the nurses. System incentives influence the behavior of the doctors, and while quality of care issues remain and are detected by the nurses, under the current Chinese hospital system, the doctors have little incentive to identify and improve the quality and safety dimensions of service delivery.

To the best of our knowledge, this is the first study of its kind comparing the experiences of doctors and nurses in relation to HPWS practice in China. Findings of this study add further evidence support in favor of the “best fit” theory in human resources management, rather than the “best practice” theory that many HPWS researchers advocate for [55]. We recommend a consideration of “best fit” of management practice at the professional level rather than at the organizational level given that there are highly diversified professional groupings in health services. It may be more appropriate to adopt a series of ‘occupational bundles’ of management practice rather than a single combination of practices equally applicable to all professional groups.

## Figures and Tables

**Table 1 ijerph-17-04379-t001:** Characteristics of doctor and nurse respondents.

Characteristics	Range	Doctor		Nurse		*p* *
		Mean	SD	Mean	SD	
Age	19–60	38.26	8.43	33.07	7.62	<0.001
Sex	Men	62.0%		2.8%		<0.001
	Women	38.0%		97.2%		
Education	Vocational/Associate degree	7.6%		48.9%		<0.001
	Bachelor degree	37.8%		46.7%		
	Postgraduate degree	54.6%		4.4%		
Department	Outpatient	38.0%		13.9%		<0.001
	Emergency	22.5%		25.0%		
	Surgical	4.8%		9.4%		
	Pediatric	5.3%		2.2%		
	Service/Administration	23.5%		42.8%		
	Others	5.9%		6.7%		
Empowerment	21–60	44.81	6.77	42.58	7.46	0.003
Commitment	15–45	30.51	4.68	28.90	4.98	0.002
Trust	8–40	30.03	6.01	30.14	5.75	0.833

* *p* values of Chi-square tests or independent samples *t* tests.

**Table 2 ijerph-17-04379-t002:** Perceptions of doctors and nurses on management practices and patient care outcomes.

Management Practices	Range of Score	Alpha	Doctor		Nurse		*p*
			Mean	SD	Mean	SD	
Job security	2–10	0.685	6.91	1.64	6.72	1.61	0.262
Recruitment	8–40	0.785	27.23	4.73	26.70	4.43	0.269
Training	8–40	0.888	27.85	5.14	27.38	6.20	0.432
Transformational leadership	7–35	0.937	26.60	5.30	26.43	4.86	0.754
Information sharing	7–35	0.677	22.66	3.70	22.26	3.64	0.288
Job quality	4–20	0.678	10.67	2.18	10.62	2.00	0.862
Teams	4–20	0.871	15.62	2.98	15.22	2.35	0.153
Patient care quality	16–130	0.823	69.16	7.456	65.98	10.34	0.001

**Table 3 ijerph-17-04379-t003:** Correlations between management practices and patient care outcomes.

Care Quality	Pearson Correlations with Management Practices						
	Security	Recruitment	Training	Transformational Leadership	Information Sharing	Job Quality	Teams
Total	0.128 *	0.229 **	0.357 **	0.405 **	0.187 **	0.312 **	0.134 *
Doctor	0.252 **	0.227 **	0.157 *	0.238 **	0.142	0.296 **	0.323 **
Nurse	0.020	0.223 **	0.484 **	0.569 **	0.212 **	0.323 **	−0.060

* Correlation is significant at the 0.05 level (2-tailed). ** Correlation is significant at the 0.01 level (2-tailed).

**Table 4 ijerph-17-04379-t004:** Effects of profession and management practice on ratings of patient care outcomes.

	*β*	*p*		*β*	*p*		*β*	*p*		*β*	*p*		*β*	*p*		*β*	*p*		*β*	*p*
**Test effects**																				
Profession (x)	0.183	0.007		0.181	0.009		0.172	0.011		0.148	0.021		0.180	0.009		0.160	0.018		0.183	0.007
Job security (y1)	−0.238	0.001	Recruitment (y2)	−0.062	0.444	Training (y3)	0.313	<0.001	Transformational leadership (y4)	0.473	<.001	Information sharing (y5)	0.226	0.002	Job quality (y6)	0.204	0.006	Teams (y8)	−0.323	<0.001
x × y1	0.114	0.080	x × y2	0.007	0.921	x × y3	−0.176	0.002	x × y4	−0.290	<0.001	x × y5	−0.166	0.016	x × y6	−0.125	0.076	x × y8	0.246	0.001
**Control variables**																				
Sex	−0.149	0.008		−0.146	0.010		−0.120	0.030		−0.105	0.049		−0.128	0.023		−0.143	0.011		−0.166	0.003
Age	0.107	0.054		0.075	0.184		0.097	0.076		0.108	0.041		0.056	0.317		0.084	0.131		0.117	0.034
Education	−0.014	0.820		−0.002	0.975		−0.040	0.531		−0.010	0.864		−0.010	0.877		−0.007	0.909		0.035	0.583
Outpatient unit	0.006	0.947		−0.024	0.804		−0.034	0.710		−0.071	0.420		−0.008	0.934		−0.046	0.621		−0.004	0.966
Emergency unit	0.082	0.369		0.063	0.506		0.063	0.481		0.023	0.795		0.036	0.698		0.027	0.770		0.117	0.204
Surgical unit	−0.098	0.152		−0.088	0.216		−0.062	0.359		−0.090	0.174		−0.085	0.225		−0.078	0.267		−0.087	0.206
Pediatric unit	−0.059	0.335		−0.065	0.296		−0.047	0.431		−0.061	0.293		−0.076	0.212		−0.069	0.262		−0.042	0.488
Logistics unit	−0.093	0.342		−0.131	0.198		−0.120	0.211		−0.134	0.154		−0.113	0.250		−0.157	0.114		−0.092	0.348
Empowerment	0.254	<0.001		0.243	<0.001		0.230	<0.001		0.198	<0.001		0.213	<0.001		0.211	<0.001		0.224	<0.001
Commitment	0.124	0.023		0.123	0.026		0.096	0.078		0.125	0.016		0.175	0.002		0.120	0.029		0.146	0.007
Trust	0.324	<0.001		0.282	<0.001		0.141	0.024		0.095	0.134		0.239	<0.001		0.223	<0.001		0.296	<0.001
**Model fit** R^2^ = 0.322		<0.001	R^2^ = 0.298		<0.001	R^2^ = 0.344		<0.001	R^2^ = 0.378		<0.001	R^2^ = 0.315		<0.001	R^2^ = 0.304		<0.001	R^2^ = 0.331		<0.001

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
