# Peer review of "Link of Patient Care Outcome to Occupational Differences in Response to Human Resource Management: A Cross-Sectional Comparative Study on Hospital Doctors and Nurses in China"

_ijerph, 2020, doi:10.3390/ijerph17124379_

Round 1

Reviewer 1 Report

Thank you for the opportunity to review this interesting article ‘Link of patient care outcome to occupational differences in response to human resource management: a cross-sectional comparative study on hospital doctors and nurses in China’ that addresses an important topic, differences between doctors and nurses in response to the human resource management practices and their associations with quality of care. The aim of the article, identifying the link of patient care outcome to occupational differences in response to human resource management is quite clearly set.

I have few comments: In the abstract, the sample size is not clear  “A total of 700 questionnaires were distributed to staff and 499 (71%) completed including 193 doctors and 180 nurses.” Who were the others that completed the questionnaire? This is more explained in the methods section, but it is not clear, if the ‘not nurses, not doctors’ completed questionnaires were analyzed or not.

Please note that the methods presented in table 4 are out of my competence.

The article is overall well written and carefully designed. The introduction and background section provides a good introduction of the topic and justifies the need for the paper. The methods section is designed carefully and well reported. The results provide clear information for the Chinese, but also global audience. The article overall provides important information for developing patient satisfaction and patient care outcomes. This makes an important contribution in the current and future healthcare. 

Thank you for the opportunity to review this interesting paper.

Author Response

We have revised the description of study sample and the data included for the final analyses in this study:

“A total of 700 questionnaires were distributed to doctors, nurses, allied health workers and managers and 499 (71%) were completed. Data were analyzed using a final sample of 193 doctors and 180 nurses”. (Page 1, Line 15-16)

Thanks for the positive feedback

Reviewer 2 Report

I congratulate the authors for thinking and conducting this research that is very relevant to the context of the management processes of professionals in the medical and nursing fields. I believe that the results of this study will have positive impacts in the context of care and positive reverberations in patient care.

Here are some suggestions for improving the article:

Introduction: It is clear, objective and with the triggering of logical ideas. I suggest making it clear at the end of the introduction of the objective of the study.

Methods: More details on the selection of recruited participants.

Author Response

Thanks for the encouragement.

We have added the objective of the study at the end of the introduction section:

“The current study aims to explore the links between patient care outcomes and occupational differences between doctors and nurses in response to human resource management practices used in Chinese hospitals.” (Page 2, Line 84-86)

We have added more details on the sampling process:

“Medical doctors, nurses, allied health workers and managers were invited to complete a paper-and-pencil questionnaire survey without supervision. Participants were required to read the informed consent letter on the cover page before proceeding to the questionnaire. Return of the questionnaire was anonymous and voluntary. Respondents could withdraw at any stage, complete only part of the questionnaire, or return an empty questionnaire to the collection box.” (Page 4, Line 145-150)

Reviewer 3 Report

This cross-sectional analysis of self-reported assessments on perceived quality of care, work climate, safety culture, teamwork, and leadership role of physicians and staff nurses reveals insightful knowledge about the differentials in  work environment viewed by two broad categories of medical professionals in two acute care hospitals in China. Although the sample size is relatively small, the paper offers useful information to guide professional training, leadership development and performance improvement.  Pertinent references are cited and discussed.

One major area could be improved, however.  The psychometric properties of measurement instruments should be detailed.  The postulated differences in perceptions of work environment between physicians and nurses could be articulated and elaborated from human resources management approaches. The limitations on the lack of validation of measurement instruments should be noted.  Furthermore, a confirmatory analysis could be executed to demonstrate the validity and reliability of measurement instruments.

Author Response

Thanks for the positive feedback.

We have added more descriptions and discussions about the reliability and validity of the measurement instruments. In the methods section, Cronbach’s alfa coefficients were added for each measurement instrument (Page 4, Line 179, 187, 193; Page 5, Line 198, 203).

We performed confirmatory factor analysis on the HPWS measurements, which was followed by sensitivity tests:

“Sensitivity tests were performed to test the robustness of the findings. The confirmation factor analysis (CFA) on the HPWS measurements with a seven-factor model showed that some items had low factor loadings (<0.4) on their respective latent variables. Deletion of these items improved and generated a good model fit. The regression analyses using variables exclusive of these items produced consistent results to those without exclusion of these items” (Page 5, Line 226-230)

In the discussion section, we discussed the limitation of the measurement instruments used in this study:

“One of the strengths of this study is the unpacking of the HPWS components, with measures derived from existing empirical studies. Unlike single measures for HPWS where a coherent theoretical framework is proposed [53], these multiple HPWS component measures have varied origins. Some of the measures may be more closely aligned with the formative assumption rather than the reflective assumption [54], in which case the classical validation methods may not be appropriate. Further HPWS measurement development is warranted, as there is no one ‘correct’ HPWS measure.” (Page 10, Line 382-387)

We have added some discussions about the role of human resources management on the professional differences:

“Human resources management approaches are likely to have contributed to the development of occupational differences in professional culture between doctors and nurses and their responses to management practices. Status distinction is highly accepted in the health industry despite concerns of its potential detrimental effects on patient safety [6,50]. Nurses have a long history of working collegially with managers and supporting effective management policies and practices in hospitals [51]. However, doctors, with more independence of practice, have been less concerned with management policies and practices. It is important to note that the rising consumer movement and the increased acceptance of clinal governance by the medical profession are likely to encourage doctors to become more responsive to management policies and practices in the future.“ (Page 10, Line 351-359)